# Multilateralism under Fire: How Public Narratives of Multilateralism and Ideals of a Border-Free World Repudiate the Populist Re-Bordering Narrative

Kesi Mahendran *, Anthony English and Sue Nieland

Faculty of Arts & Social Sciences, School of Psychology & Counselling, The Open University,
Milton Keynes MK7 6AA, UK; anthony.english1@open.ac.uk (A.E.); sue.nieland@open.ac.uk (S.N.)
* Correspondence: kesi.mahendran@open.ac.uk

**Abstract:** How do global multilateral arrangements such as the United Nations (UN) and World Health Organization (WHO), vital to post-pandemic recovery, connect to the public understanding of multilateralism? The Citizen Worldview Mapping Project (CWMP) conducted in England, Scotland and Sweden examines how the degree of migration–mobility interacts with worldviews. CWMP asked participants (N = 24) to rule the world using an online interactive world mapping tool. Citizens were first interviewed on their migration–mobility, then invited to draw or remove borders on the world to manage human mobility. Citizens then engaged in a dialogue with António Guterres' 2018 address to the United Nations General Assembly on multilateralism. Dialogical analysis showed how, when empowered to rule the world, the majority of participants, irrespective of the degree of migration–mobility, expressed an ideal of a border-free world, even if they then went on to construct borders around the world. We understand this as a democratic dialogical ideal of a border-free world. Participants articulated rich narratives and social representations of international relations, yet did not have a formal understanding of the reified concept of multilateralism. Bridging this gap between the *consensual sphere* of the public's ideals based on social representations of cooperation and conflict and the *reified sphere* containing political narratives of multilateralism is a key step to longer-term post-pandemic recovery. A first step will be further studies into how an ideal of a border-free world can reconfigure political resistance to xenophobic populist re-bordering.

**Keywords:** multilateralism; migration; political narratives; dialogical self; European Union; one world; global identification



## 1. Introduction—New Nationalism and Populist Re-Bordering beyond COVID-19

It is an attractive starting point, within the context of this Special Issue's concerns with how the pandemic has reconfigured political resistance, to propose that the rupture of the COVID-19 (Coronavirus) pandemic has created emergent forms of governance rooted in cross-border solidarity, that state–civil–society arrangements acutely aware of our global interdependency changed towards new forms of togetherness. The UN, WHO and European Union all proposed such a grand narrative, that we are all in it together, appealing to states and citizens to act together through multilateralism (United Nations 2020). A risk with using the seemingly cohesive concept of 'togetherness' to harness certain civil behaviours from the public is the narrative that we are *all in it together* contains within it a self-reliant ideal of resilience (Müller and Tuitjert 2022; van Uden and van Houtum 2020). To this end, as political psychologists preoccupied with the dialogue between citizens and their governments, we propose an alternative departure point: that the capacity for cross-border solidarities and global-level cooperation was already emerging as a repudiation of widespread populist re-bordering. The field of decolonial and postcolonial studies offers valuable insight here on the focus of populist leaders' narratives and how they are imbued with nostalgia and nationalism (Campanella and Dassù 2019; Koegler et al.

2020). However, the political psychological focus of this paper is on the public themselves and how their sense of global solidarities resisted the active mobilisation of polarising narratives such as Protecting European Values (EU), Make America Great Again (US), Take Back Control (UK), Stop The Boats (UK) and *Folkhemmet* (People's Home[1], Sweden) These public narratives emerged to refuse, resist and counteract the polarising narratives integral to new nationalism. New nationalism has arisen as a concept to articulate the mainstreaming of radical right-wing parties into parliament coalitions platformed on law and order, anti-immigration policies and, often, anti-European Union sentiment (Eger and Valdez 2015; Korkut 2020; Bitonti et al. 2022). New nationalism builds on fantasy border narratives built around the myth of a homogeneous nation (Kinnvall and Singh 2022) and resists alternative futures (Krasteva 2020; Yerly 2022; McAuley and Nesbitt-Larking 2022).

Drawing on the Citizens Worldviews Mapping Project (conducted in 2019), we demonstrate that citizens, when invited to rule the world within an online interactive worldview mapping task, express before the coronavirus pandemic and the invasion of Ukraine an ideal of a border-free world and articulate narratives of togetherness and global interdependencies. Exploring the dimensions of these narratives provides the potential to understand the solidarities citizens freely express when discussing human migration and mobility, the potential to understand how citizens would configure the global order when they are engaged in world-making (Power et al. 2023) and the potential to understand everyday resistance to *business as usual* and other post-pandemic hegemonic narratives (Andrews et al. 2023).

The dialogical narrative analysis (Bakhtin 2010; Marková 2003; Fathi 2013; Mahendran et al. 2022; Mahendran et al. 2023) below works with three individual cases to ask one central question: *How does the ideal of a border-free world connect to the public's articulation of multilateralism*? In order to examine this question, the analysis interrelates two parts of the Citizen Worldview Mapping Project (CWMP): first, how citizens mapped the world and the democratic dialogical ideals they expressed when doing this, and second, how they responded to António Guterres' statement in his 2018 speech, that multilateralism was under fire.

Central to the theory of social representations is Moscovici's delineation between the reified universe and the consensual universe. The reified universe of science creates seemingly ahistorical, objectified scientific knowledge, and the consensual universes of common sense meanings involves a thinking public elaborating on unfamiliar concepts, objectifying them and anchoring them to other, familiar knowledge (Moscovici 1984; Moscovici 1988; Howarth 2006; Mahendran 2018; Mahendran et al. 2022). This articulation is sustained through intersubjective negotiation (Gillespie and Cornish 2010). One striking finding of CWMP is that none of the participants understood the formal or reified term *multilateralism*. Multilateralism, as we demonstrate below, is a concept that needs to be translated and anchored into people's common sense worlds. Public understanding of the ideals of actors such as the UN therefore requires a bridge between multilateralism's formal articulation within global governance and what other concepts the public connect multilateralism to in order to make sense of it. Exploring the public's worldviews on the global order, nationalism and transnationalism are the very foundational piers to this bridge. The abutment to the bridge, we propose, is their ideal of a bordered or a border-free world.

Using the concept of worldviews implies perspective or conception of the world, and it is important throughout to consider the extent to which consciousness has shifted toward a more planetary consciousness (Chakrabarty 2019). A challenge for social and political psychology is that, given the extent of our global interdependencies, scientists need to go beyond psychology's long-standing preoccupation with worldviews or worldmaking and instead consider post-human/more-than–human planetary consciousness (Chakrabarty 2019; Haraway 2016; Mahendran et al. 2022). We have designed methods and tools to support citizens in articulating *planet views* that recognise the climate emergency. The study presented below offers an analysis of new psychologies of multilateralism and international relations, which recognises the role of national sovereignties in tackling the climate emergency. The remainder of the introduction develops the necessary bridge by

exploring three components: (i) reified articulations of multilateralism, (ii) the existing literature on the public's understanding of multilateralism and (iii) studies into global human identification and citizenship.

## 2. Multilateralism and António Guterres' Addresses to the UN

Multilateralism—the collective consideration of global matters by world nations—is a relatively new concept connected to post-war arrangements around 1919 (Alhashimi et al. 2021). Multilateralism, in its present form, is generally considered to have emerged in the aftermath of the Second World War and diminished League of Nations (Schlesinger 2003). The broad purpose of multilateralism is to institutionalise intergovernmental cooperation and achieve common goals, typically within organisations such as the United Nations or G20 (Langenhove 2010). One of the more commonly used traditional definitions is Keohane's (1988) assertion that multilateral agreements are optional endeavours which offer a 'persistent sets of rules that constrain activity, shape expectations and prescribe roles' (Keohane 1988, p. x). More recent thinking on multilateralism highlights the ever-growing necessary, rather than optional, status of multilateral cooperation, given the growing importance of global public policy on climate-based threats (Kaul 2020).

In 2021, as the world was engaged in cross-border cooperation to develop a COVID-19 vaccine, António Guterres closed his annual address to the United Nations proposing:

> The best way to advance the interests of one's own citizens is by advancing the interests of our common future. Interdependence is the logic of the 21st century. And it is the lodestar of the United Nations. This is our time. A moment for transformation. An era to re-ignite multilateralism. An age of possibilities. Let us restore trust. Let us inspire hope. Address to the General Assembly of the United Nations. (António Guterres, September 2021)

At the time of designing the CWMP in 2018, just over a year before the COVID-19 pandemic, within the context of rising populism, polarisation, antipolitics and antidemocracy, António Guterres had used his address to emphasise the extent to which multilateralism was under fire. He stated:

> The world is more connected, yet societies are becoming more fragmented. Challenges are growing outward, while many people are turning inward. Multilateralism is under fire precisely when we need it most. Address to the General Assembly of the United Nations. (António Guterres, September 2018)

In his September 2022 UN address, Guterres did not go further into how we might 're-ignite multilateralism', but the connotations around the expression multilateralism under fire radically altered. Russia's invasion in February 2022 of Ukraine led to a radical alteration across Europe of the country's relationship with NATO and the European Union. Ukraine sought to fast-track its acceptance into the EU. Equally, both Finland and Sweden took steps to join NATO. Guterres took the Black Sea Grain Initiative as an example of 'multilateral diplomacy in action', as Türkiye, Russia and Ukraine agreed to a grain arrangement. His rallying cry at the end of his speech foregrounded such cooperation and dialogue:

> At one stage, international relations seemed to be moving toward a G-2 world; now we risk ending up with G-nothing. No cooperation. No dialogue. No collective problem solving. But the reality is that we live in a world where the logic of cooperation and dialogue is the only path forward. No power or group alone can call the shots. No major global challenge can be solved by a coalition of the willing. We need a coalition of the world. Address to the General Assembly of the United Nations. (António Guterres, September 2022)

If supranational organisations such as the European Union, NATO and the UN rely within liberal democracies on a public mandate (they rely on public assent to multilateralism) how does this continue without public dialogue and public understanding of multilateralism? Surveys show that, when asked about trust in governments, citizens tend to have high levels of trust in the United Nations (Eurobarometer), higher than their trust

in their national governments. Therefore, a lack of recognition of the concept of multilateralism itself should not be taken as a lack of recognition of the role of the United Nations.

## 3. Public Understanding and Attitudes towards Multilateralism

We found no literature on the public's understanding of multilateralism or the public's views on global governance. Questions related to global governance are not asked within social attitude surveys or within psychological studies. Yet, as this article argues, the understanding of institutions such as the United Nations, World Health Organization, World Bank and the core concept of multilateralism are key to post-pandemic recovery and cross-border cooperation. Scholarly examinations into multilateralism have tended to concentrate on state-level analyses of multilateralism within the context of post-Cold War polarity debates and the ongoing role of global governance and international relations arrangements, such as the UN and NATO.

Examining attitudes towards multilateralism and how this relates to the public understanding of international relations could be insightful for a better interpretation of public responses to new nationalism and populist re-bordering. Whilst the British Social Attitudes Survey has a variety of data on national identity and the UK's EU relationship (British Social Attitudes Survey 2022), there are no questions relating to multilateralism or its related organisations (e.g., the UN). Equally, the survey data on public attitudes across various European countries, such as Eurobarometer, European Quality of Life Surveys and European Social Survey, does not measure public attitudes towards multilateralism.

## 4. Global Identification and Citizenship

Perhaps the closest line of inquiry is to be found within social and political psychology with the growing interest in global identification and citizenship (GHIC) (McFarland et al. 2019; Mahendran et al. 2023). Of importance to gauging public understanding of multilateralism is psychology's existing long-standing interest in global consciousness. Sampson and Smith in the 1950s gauged the extent of people's agreement with the statement 'it would be better to be a citizen of the world than of any particular country' within their Worldmindedness Scale (Sampson and Smith 1957; McFarland et al. 2019; Mahendran et al. 2022). A landmark departure point in this field is Sam McFarland's essay 'The slow creation of Humanity' (McFarland 2011). Within political psychology, different terms have been favoured, e.g., identification with all humanity (IWAH-McFarland 2011; McFarland et al. 2012), global citizenship identification (Reysen and Katzarska-Miller 2013), the global social identity scale (Reese et al. 2014) and psychological sense of global community (Hackett et al. 2015). When reviewing this line of inquiry, McFarland proposed the term global human identification and citizenship (GHIC) (McFarland et al. 2019). However, whilst various GHIC scales ask questions on global identification, they do not ask citizens to talk about global orders, cooperation or, crucially, the concept of multilateralism.

Psychologist Fathali Moghaddam (2020) argue that psychologists have a substantial role in achieving democracy across borders by understanding democratic citizenship at the level of the individual. Moghaddam proposes that, to strengthen multilateralism, psychology should focus on 'omniculturalism'. Indeed, he argues that an educational policy based on this concept would focus on emphasising human commonalities rather than exacerbating national or group differences (Moghaddam 2012). In essence, he concludes, a universal category of 'human being' allows all to adopt this as a superordinate identity and, thus, move beyond intergroup conflict. Bilewicz and Bilewicz (2012) argue that defining universal human traits in the first instance is a problematic concept that is unlikely to be perceived as legitimate by all groups. Moreover, research on superordinate identities within the context of national identities often has limitations regarding intergroup projections (Kessler et al. 2010). That is to say, any higher-level social categories are defined by the public's knowledge of their own subgroups (Wenzel et al. 2008). Therefore, any universal definition of humanity is unlikely to align with this assumed knowledge due to the disparate nature of group membership.

Another line of enquiry within psychology literature in the context of public understanding of multilateralism is as an explanatory tool for why US policymakers opt for multilateral solutions in response to international security threats (Neack 2013). For example, Rathburn's (2012) proposal that social psychology research on generalised trust offers a paradigm for understanding that cooperation follows trust, rather than vice versa. An issue here is that focusing on either superordinate identities or generalised trust does not offer any insight into the public's engagement with the concept of multilateralism. Indeed, there seems to be a vacuum in the social scientific literature on the public's narratives or understanding of multilateral cooperation; specifically, the tension between a desire for security and sovereignty (i.e., the endorsement of borders) alongside a wish to participate in global cooperation beyond the limitations of borders.

As we have argued (Mahendran et al. 2023), a difficulty with studies into global identification is that, across the different measures, respondents showing global identification remains strikingly low. There have been some attempts to use the climate emergency to increase it, but these have had limited success. We propose that this is because such measures tend not to grapple sufficiently with migration–mobility and precarity. Precarity is a key concept for social psychology (Coultas et al. 2023; Fine 2023; Mahendran et al. 2023), and within the design of the study presented below, we measured participants' degree of migration–mobility using the Migration–Mobility Continuum (Mahendran 2013).

## 5. The Present Study—Dialogical Citizens

Citizens who are navigating and making sense of an increasingly politically turbulent world often draw upon stories and narratives that are prevalent in society, for example, associated with conflict, change, gender, culture and security (Andrews et al. 2015; Hammack and Pilecki 2012; Nesbitt-Larking 2022). These stories relate to the predominant social representations that come and go throughout life and are often reflected and seized upon when citizens are brought into dialogue with the political world they inhabit (Mahendran et al. 2015; Zittoun 2017).

These can be contemporary or historical, so they may be part of an autobiography that reflects the past and the present and are reflected by reference to sociopolitical events, as well as personal events. They are likely to reflect changing political social representations that run alongside a person's story and become explicit in narratives that focus upon salient social and political events, for example, responses to the UK's decision to leave the EU and Brexit (Mahendran 2018; O'Dwyer 2020), to immigration (de Rosa et al. 2021) or to the US political division between left and right (Hanson et al. 2021). They emerge when researchers bring their participants into dialogue with the predominant political narratives that influence contemporary thinking; otherwise, they may be unrecognised and subsumed into the view of narratives as only autobiographical.

The present study facilitates the articulation of political narratives by bringing participants into dialogue within macro-level narratives, using stimulus materials such as films, speeches, images and governance policy statement on vexed issues. This design allows participants to reflect in their narrative how these questions are, and can be, addressed and answered within the key available social representations. Using the four-step analysis below, it is possible to explore one world narratives (Mahendran 2017), narratives of bordering and its relationship to precarity (Mahendran et al. 2023), de-polarisation through sustaining dialogue (English and Mahendran 2021) and resistance to nostalgia rhetoric (Nieland et al. 2022). These materials, alongside the Migration–Mobility Continuum (MMC; Mahendran 2013), allow us to reveal the relationship between the understanding and appreciation of multilateralism and parameters of human mobility.

**6. Methodology**

The present study combines two methods: face-to-face interviews and an online interactive worldview mapping tool (IWMT), which was developed by interactive media developers Ryan Hayle and Kesi Mahendran. Social scientists have made considerable use of maps in order to access worldviews (Futch and Fine 2013; see Fine 2023 for US children's maps of their school environments). In our design, rather than explore psychosocial mapping and sense-making, participants are brought into direct dialogue with images of the Earth and then stimulus materials such as factual questions and political speeches on international relations, as outlined above. This dialogical design draws upon Bakhtin's concept of the dialogical self and social representations (Mahendran et al. 2022, 2023) to understand how participants co-author key political concepts such as multilateralism. Participants can be understood as taking up the *I-citizen position*. This citizen position (Mahendran et al. 2015) arises out of an Arendtian notion of 'enlarged mentality' (Arendt 1961), where participants think beyond their immediate interests—they think cooperatively (see also Dewey [1927] 1954 on public capacity).

Within deliberation studies, there is some debate as to how and particularly *where* citizens articulate their interests in terms of rational speech and using formalised political discourse. Seyla Benhabib, in defending Arendt from critique by the feminist such as Adrienne Riche, that Arendt took a masculinist view of public dialogue and drew on Arendt's study of Rahel Varnhagen. This explores the salon space as a space for playful, risky talk often avant-garde and quite distinct from the formalised political discourse (Benhabib 1995). The disinhibitions of salon talk could be the key to truly understanding citizens as dialogical citizens in the context of social media and its potential.

*6.1. Sampling Participants and Our Positionality*

Fieldwork was conducted in 2019 in Edinburgh (N = 10), Stockholm (N = 10) and Manchester (N = 3). The Manchester component was halted in February 2020 because of the COVID-19 pandemic. The study involved 11 males and 12 females, and the ages ranged from 19 to 69 years old (M = 41.18). The mean age was altered slightly between cities as follows: Edinburgh, M = 46.33; Manchester, also M = 46.33; and Stockholm, M = 35. Quota sampling across the degree of migration–mobility (blinded/see above) was used involving an initial discussion to establish their degree of personal migration–mobility. Adverts on online neighbourhood sites and a notice at Stockholm University were the key two steps to create the sample, followed by chain sampling. Participants came from professional, skilled and semi-skilled occupations. There were two academics, three students and no unemployed people in the sample. Both the interviewers are British, the first author (MMC2) has parents who were migrants from Sri Lanka. She conducted a set of interviews at all three locations. The second interviewer (MMC4) worked only in Scotland. She moved from England to Scotland, having spent a year working elsewhere in Europe. The extracts below are presented as dialogue to support further reflexive reading of the analysis presented.

*6.2. Procedure*

Interview: The semi-structured interviews were on, average, 35 min long. The interviews opened with sentence completion questions: 'The world is...', 'I am a part of...' and I vote/don't vote because...'. Participants then answered questions on citizenship, including the question 'Do you consider yourself a citizen of the European Union?'. In the next section of the interview, participants answered six questions that enabled the authors to place them within one of the ten positions within the Migration-Mobility Continuum (see Figure 1). These six questions asked about the moves the participant had made, whether they had moved and returned to the country of the interview, whether they had ever planned to move, whether they felt on the outside and, finally, whether they were settled or would move again/for the first time (Mahendran 2017).

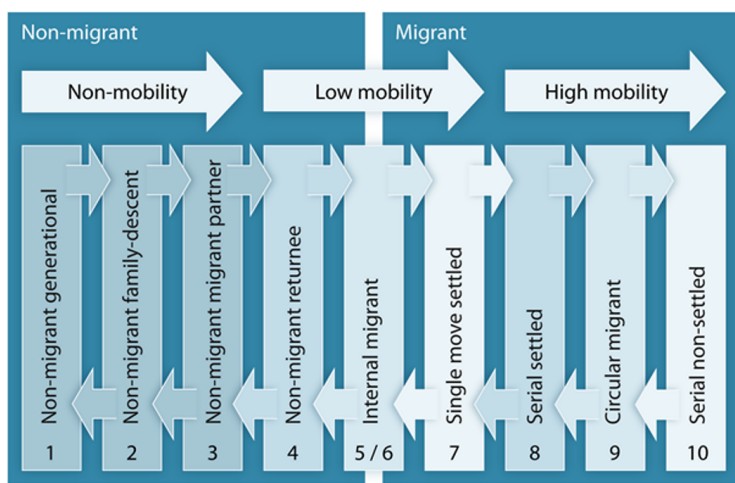

**Figure 1.** The 10-point Migration–Mobility Continuum (Mahendran 2013, 2017).

*Interactive Worldview Mapping Tool (IWMT)*: The duration of the mapping was 60 min on average. This included a break between the interview and IWMT mapping. The total session for the study as a whole was 90 min mean average (range 78–108 min). The Interactive Worldview Mapping Tool (IWMT) involved four sections. Section 1 involved participants responding to three open questions and the same six MMC questions they had answered in the interview. These were now presented as closed drop-down options, in order to explore the losses and gains of quantification in future studies. In Section 2, participants could choose between two map options (Figure 2). Participants were invited to choose which they preferred and were told that they now had the power to *rule the world*. Both maps are based on the widely used but contested Google Earth's Spherical Normal (equatorial) variant of the Mercator projection. Participants then saw the statement:

> *Draw lines around the parts of the world that you feel require state lines. Each time you draw a line on the map-this represents a boundary where people travelling across the boundary would need to show their passport to enter/or be attempting to claim asylum.*

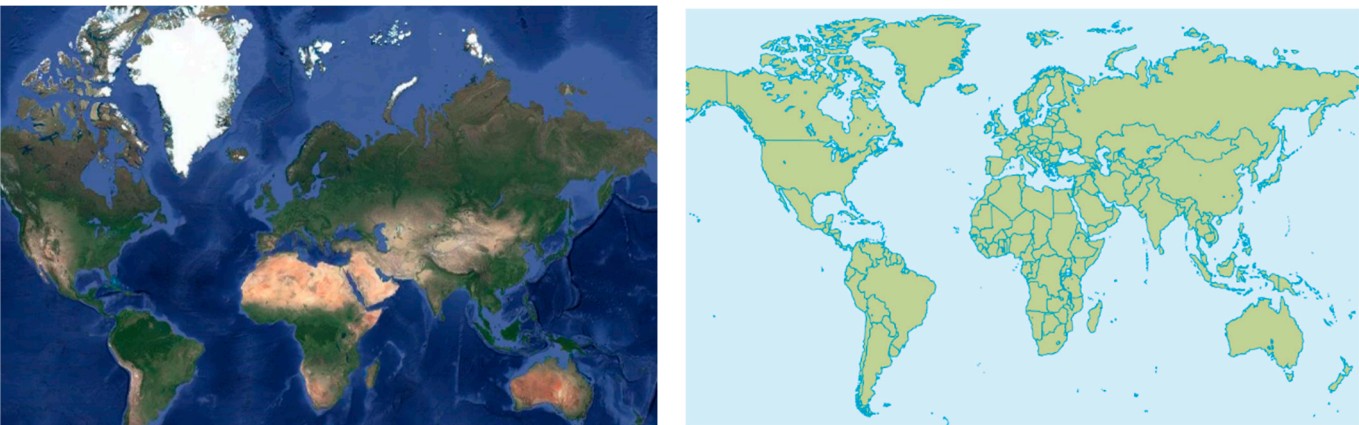

**Figure 2.** Two options: border-free Earth map and nation state borders world map.

Participants were given as much time as they required to complete and could switch maps. Section 3 involved participants answering ten international relations questions. Eight were closed-response factual questions, e.g., 'Indicate which countries began the European Economic Community in 1951, click on the list which accurately shows the ten wealthiest countries (nominal GDP); click on the countries which were part of the Swedish/British Empire at its height in 1648/1922'. Participants then saw the correct answer. However, the final two questions were not factual but attitudinal questions assessing their agreement with two statements. Statement 1 was taken from Jean Claude

Juncker's (President, European Commission) State of the Union address. This began 'I want Europe to get off the side-lines of world affairs'. Statement 2 directly refer to the title of the article *Multilateralism under fire* by António Guterres (Secretary General United Nations), as set out above. Finally, in Section 4, participants were given a second opportunity to re-draw their worldview map. The hypothesis here was that, after being confronted with the parameters of their international relations knowledge, they might moderate their actions when ruling the world.

### 6.3. Analytical Steps

Both the interview and mapping were conducted in English, this was then transcribed and built into a database using NVivo 12 by the second author. Within the approach, we have developed within the Public Dialogue Psychology Collaboratory (PDPC), the analysis moves iteratively between four steps (Mahendran et al. 2022). When conducting the dialogical analysis of the maps and the recorded interviews with the MMC position, it became evident that participants were working with an ideal of a border-free world. This related to differing sense-making on issues of control and sovereignty, as well as distinct social representations about human agency and the potential role of borders (Mahendran et al. 2023). In the third step, using NVivo 12, we analysed all the responses to Statement 2 made by Guterres. This was understood by the authors using three underlying social representations about how the world is organised as conflict-based, competitive or collaborative/cooperative (Staerklé et al. 2011; Mahendran et al. 2023). In the fourth step, key I-positions within the dialogical self were identified within the transcribed interview dialogue. In the analysis presented below, we focus on three cases where there are low levels of migration–mobility and the participants use their national identity to examine transnationalism and then multilateralism. Though not articulated here, minor transnationalism appears key to bridging the gap between reified and consensual understandings of multilateralism.

### 7. Dialogical Analysis

It is important to keep in mind the timing of the fieldwork (2019) occurred before the COVID-19 pandemic and the Russian invasion into Ukraine. Multilateralism under fire takes on a new meaning within this context, and the decision by both Sweden and Finland to apply for membership to NATO may well create new narratives amongst the public on questions of multilateralism. As discussed above, in order to reveal the features of the public's understanding of multilateralism, the contribution we make within this article is to relate this to an ideal of a border-free world and the decision to control and remove borders. The three cases selected are presented as a dialogue between the citizen and the first author. They are presented in the following order, which relates directly to the procedure of the study. First, the participants' decision to control/remove borders on the world, then their response to the Guterres' address, followed by their decision to control/remove borders.

Each case shows how participants use their hyperagentic position differently. In Cases one and three, agency is about making careful decisions on how to border the world. The second case agency reveals processes of indecision. Our focus here is not on whether or not participants control/remove borders but rather on how participants articulate the ideal of a border-free world and how this relates to multilateralism.

### 7.1. Case 1: YD—Stockholm

In the first case, YD articulates an ideal of a border-free world before receiving any stimulus material. He works with the border-free Google Earth map. Whilst he could work with a state-bordered world, he chooses to define where supranational borders should be, relating to cooperation between certain regions. YD is one of the youngest participants, a solider aged 20, who has never moved from Sweden and who describes himself as 'fully settled' (MMC1). He moves between a series of I-positions. Within YD's ideals of no borders, the concept of the 'Earth' is used. He does not refer to the world, planet or globe; instead, the Earth is figured as containing 'united states' all working



cooperatively together. When referring to 'states', it is important to note that 'borderless' means working across borders rather than borders not existing. Within this narrative, state entities work together to advance beyond the parameters of the Earth, 'going to space'; this is understood as productive. JD's understanding of international relations is a so-called realist one: states killing each other, which builds on an antinomy of productive/destructive social representation.

### 7.1.1. Extract 1—United States of Earth

> **First Author:** Which is the world that you, you see, you know, the, the one that you would see the world as, you know, being like or how you would like it to be.
>
> **YD:** How I would like it, want it to be? Uh-huh. Ideally, I'd like, like, um, United States of Earth kind of thing. Instead of, instead of killing each other, we can actually do productive stuff like, I don't know, going to space or something. So I'd like to see this world map, ah, the borderless world one, but most realistically is another question. (YD, Interview, MMC1, Stockholm).

YD, in his statement within the IWMT (Extract 2), introduces an I-worker position, to imagine economic migrants moving across the world. This social representation of the world divides it along what might be understood as *international developmental* lines, and the expression 'moves up' evokes a representation of a global north/south divide. Finally, having introduced the idea of conflictual cultures, YD makes the decision to include Russia to create cooperative diplomatic international relations. YD spent around 10 min creating his borders, and Figure 2 shows the care taken around where to place the lines in his final map.

### 7.1.2. Extract 2—The Distribution of Wealth

> **YD:** I thought about distribution of wealth and the expected flow of population. I put the EU and Russia in the same box since all of them are wealthy and well-developed countries, and within the EU the ideals are somewhat similar. I included Russia to minimize hostility between the regions. I then boxed Africa and the way I see it EU would be responsible for economic stimulation of Africa. Developing infrastructure in Africa as well as helping establish working democratic governments. After that I made the same argument with USA/Canada and Latin America. Oceania is one region due to the shared island property as well as the economic power and well-developed status of Japan and Australia would allow them to stimulate the other countries in their region.
>
> China/far east region was the most difficult due to China's very particular culture compared to the other countries in the region, but I think the economic power of China and India would allow them to be responsible for development of the other countries in the region. The Oceanian region could support economically as well (YD MMC1-IWMT Statement)

Within this statement, YD explains that the ideals between the EU and Russia are 'somewhat similar', which points to a 2019 pre-Ukraine context, though this overlooks the Annexation of Crimea that occurred in 2014. YD, placing both within the same zone, minimises hostility. The EU (a geopolitical entity) is placed in the position of being 'responsible' for Africa. Africa, strikingly, is not understood as a geopolitical entity or a set of heterogeneous countries, despite the existence of the African Union and trans-African initiatives. Africa is represented in deficit (lacking infrastructure and democracy as an entire region), and the same colonial arrangements are set up with the USA/Canada and Latin America. The social representational understanding of international relations is that there are economically powerful countries who will be responsible for the other countries. In Extract 3, YD dialogues with Guterres' statement by returning to his opening statement (Extract 1).

### 7.1.3. Extract 3—We're Cooperating Less

> **First Author:** this time it is the United Nations. So, he said a slightly smaller statement. The world is more connected, yet societies are becoming more fragmented. Challenges are growing outward, while many people are turning inward. Multilateralism is under fire precisely when we need it most. So, multilateralism in the sense of countries working together (YD: Mm-hmm) What do you think of that statement?
>
> **YD:** I have to agree because, um, like I said in my, in my opening, actually the first 10 min, uh, I don't like how, uh, nationalism is growing stronger in different countries (First author: Yeah) We're cooperating less and less. Uh, it's becoming more important for like nationalistic parliaments are winning around in all countries.
>
> **First Author:** Yeah. Yes, yes, you started on that note, didn't you? Yeah. Yeah, it's true. So, what do you want to write in then?
>
> **YD:** Okay, I agree the world is becoming more fragmented. Nationalism is on the rise everywhere, and cooperation is decreasing. The threat of war is increasing. Countries are increasing their defense budgets.
>
> YD when given the opportunity to revise his map after responding to the set of IR questions explains 'I think the borderless one's more fun to look at'. He adds that he is 'fairly satisfied' with his original mapping (Figure 3 and does not alter it (YD, Interview, MMC1, Stockholm).

YD's understanding of multilateralism is based on cooperation—if we, as nations, cooperate, we would not need to increase defence budgets. Yet, returning to the neocolonial context, we have connected to rising new nationalism, and the basis of cooperation is organised around hierarchies of leader-developed countries and follower-developing countries. Here, multilateralism is related to economic productivity and exploration. Multilateralism becomes the basis of the exploitation of resources. Therefore, whilst YD may be pro-multilateralism in a potential social attitudes question within a survey, the qualitative narratives around 'alliance' that inform this decision are key. YD's worldview can be contrasted with the second case: PR. PR selects the Google Earth border-free world. However, he makes the decision not to place any borders on the Earth at all (Figure 4).

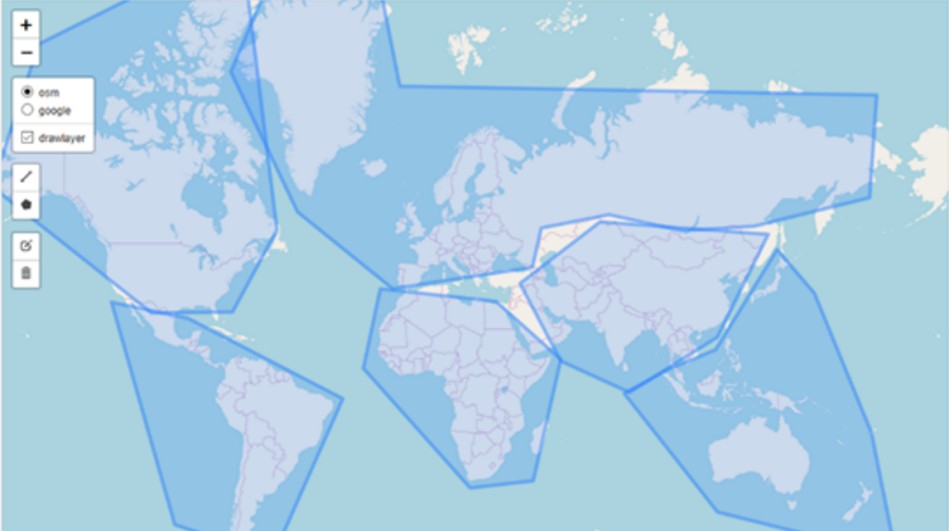

**Figure 3.** YD creates a world with regional borders.

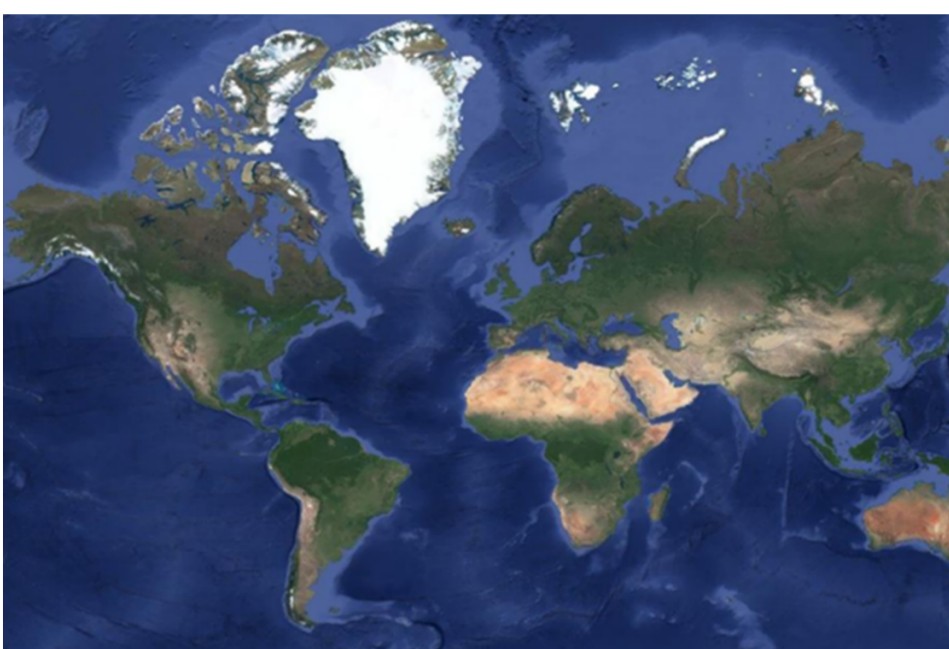

**Figure 4.** PR-preferred map.

*7.2. Case 2 PR—Edinburgh—Border-Free World*

7.2.1. Extract 4—It Is Not for Me to Place Barriers between People

> **First Author:** So, if you ruled the world PR how would you border the world?
>
> **PR:** I suppose (.) I wouldn't, or if I did, I wouldn't presume I wouldn't be able to do it (.) I can't make that choice, sorry there is no way I can put a line across anybody, it is not for me to say, it's just not for me to say, so (.) (First author: Tell me a bit about that) Erm (.) so fundamentally I don't think there is any difference between people living in Amsterdam or Manchester, they speak a different language so I'd either say the countries as they are is fine, basically that's fine, countries as they are, its fine, it's come through a historical process, that's fine, I'm not going to disagree with it people are largely happy with those things, but I couldn't, I couldn't (pause) I'd find it upsetting to think I'd be stopping people moving from one place to another, I couldn't do it.
>
> **First author:** So, would you keep the world like that, or would you keep it like the state bordered one?
>
> **PR:** (several seconds pause). I'd refuse to have any agency in that decision making, on my own, fundamentally, there is no, there is no, any ideal in my head, but I wouldn't want to be responsible. So let's assume I am the king of the world or president of the world and we say it would be useful to have some sort of control and there was really good reasons for that and I accepted that, those sorts of checks or controls, the process to finding what they ought to be, would not be my agency it would have to be decided by something else by the people who live there. For good reasons so erm (.) yeah, who am I to say who could live here or can't live here, it is for us to say, do you know what I mean, but it's not for me to say (PR, Interview, MMC5, Edinburgh).

PR foregrounds his 'individual agency' as not being enough to make the decision on whether or where borders should be placed. Yet, barriers are understood as existing potentially for 'good reasons'. PR's no bordering position is not an ideal for border-free worlds; rather, it translates borders into 'barriers' and understands the world as being populated by people. Within the statement PR writes within the IWMT, he encapsulates his I-position around individual agency.

**PR:** I drew no lines because, it is not for me to place barriers between people. Those barriers may exist, and for good reason, but my individual agency should not be the determining factor (PR, IWMT statement, MMC5, Edinburgh).

This is quite distinct from YD, who takes up the position of the state actor. PR's response to Guterres' statement further develops his understanding of multilateralism and how this relates to decisions to control/remove borders on the world.

### 7.2.2. Extract 5—I Would rather a Word like Cooperation

**First author:** So, this is Guterres now. "The world is more connected. Yet societies are becoming more fragmented. Challenges are growing outward while many people are turning inwards. Multilateralism is under fire precisely when we need it most." What do you think about that statement?

**PR:** It probably lies with who we are (...) I think you could maybe particularly that France is more fragmented than it was 50 years ago. Do you know what I mean? (First Author: Yeah) The normal way of understanding what it was like to be (...) on the other side of the world, that's not quite true, they did have a little bit more (...) But yeah generally, a sentiment I agree with. I wouldn't use the word multilateralism but.

**First Author:** You say you wouldn't use the word multilateralism, tell me a little bit about why you wouldn't use that term.

**PR:** Um. Probably aesthetic reasons actually. I would rather a word like cooperation, or ... so the reason I don't like that I guess is because that's talking at state access as if they're completely different to people (First Author: State access as if they're completely different to people). Yeah, so the relationship between States, and it is, it is different, and I, I'm not going to argue that they're not, but you start using words like that that have absolutely no basis on people's lives. It sounds like a foreign, it sounds foreign, I think it sounds, it's a foreign word for dealing with foreigners. Whereas if you say, actually, looking after each other is how we come into common solutions, working together. It's words, common words that people understand that relate to how they work in their gardens or volunteer at a bowling club, that works. And it's the same, it's the same underlying idea that working together allows you to create stuff, outcomes that you might like that you can't achieve on your own, you can't have bowling games on your own, it doesn't work (PR, Interview, MMC5, Edinburgh).

PR takes up an advisory position between the state actor and the people saying 'if you start using words like that, the concept is not going to be understood'. He proposes that it is not a question of the idea of cooperation or alliance; rather, it is about people's lived realities. PR proposes a more social relational public narrative of multilateralism, which privileges 'looking after each other' and 'common solutions'. The image of citizens bowling evokes the idea of Bowling Alone, possibly a reference to Putman's statements on social capital. Again, like YD, working together is about 'creating stuff', i.e., productivity, rather than peace. Citizens here are figured or portrayed as being at home, at leisure bowling, which limits the scope of their political agency. They are not portrayed as at work or engaged in sociopolitical activities, e.g., voting. In his statement within the IWMT, he points to Guterres' conflation between people and governments.

### 7.2.3. Extract 6—Guterres Equates People with Governments

**PR:** I agree that the co-operation between states to meet challenges is key. I think Guterres over-estimates the level of outward looking societies previously. There has perhaps been a growth of populism since 2008, but whether there has been a fundamental change in people's attitudes is unclear. He seems to equate people with the governments and discourse in their countries (PR, MMC5, IWMT Statement in response to Guterres' statement).

Central to PR's understanding of multilateralism is the idea that, as long as it is about governments and states, people are not going to identify with it. Here, the demos are presented as apolitical and not identified with nations, countries or speaking on behalf of their countries. Within the final case, OU in Stockholm chooses the state-bordered map (Figure 5), dividing the world into 'good' and 'bad' countries. She explains her rationale and reveals a public narrative of multilateralism around the push and pull factors of people moving from 'bad' to 'good' countries.

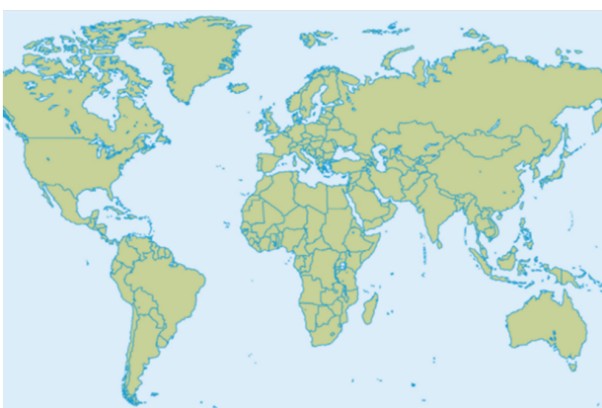

**Figure 5.** OU's preferred map.

### 7.3. Case 3—OU—Stockholm

### 7.3.1. Extract 7—The World Isn't a Perfect Place

> **OU:** Because the world isn't a perfect place, and a lot of countries don't take care of their citizens, so people feel the need to flee. Instead the countries should take care of their people, so they don't feel the need to flee. If we didn't have borders, then everyone obviously would like to live in the best countries, like Sweden, which has great social security systems. If everyone came here, then Sweden would be destroyed, so to speak. It's really hard, even now, to find a job and a place to live, for people who already live here. we need to make it better for our own people first, before we can help others (OU, MMC4 Statement on IWMT on mapping).

OU creates a state-bordered world that contains people on the move, people fleeing. She uses the term flee and equally talks of Sweden as being 'destroyed'. Her use of 'we' is about Swedish people who need to be placed first. The movement of people across the world is understood entirely in terms of refugee-related movement rather than economic movement, travel or tourism. It is important to note that, like the other participants, OU was not familiar with the term multilateralism and immediately asked what it meant.

### 7.3.2. Extract 8—Multilateralism as Fleeing People in a Connected World

> **First author:** And then we're going to go a little bit bigger now. So, the same time, i.e., September 2018, now this is Guterres.

> **OU:** What is multilateralism?

> **First author:** Countries working together.

> **OU:** Oh, okay. Thank you. Yeah. I guess one of the problems of the world being more connected is that you know a lot of people, moving around fleeing their countries and in that way creating problems in the so-called better countries. Again, Sweden is the example from here. It used to be a safe country, safer. Now, because of the high immigration, there is a lot more violence. It's a lot more insecure. So, I'm not saying it would have been perfect if we didn't allow immigration. Obviously, there are bad people within you know bad Swedish

people too disregarding ethnicity because you can still be Swedish even if you're not ethnically Swedish. It depends on how you feel, were you born here, grew up here, maybe you're born in another country but you grew up here, et cetera you know and there's many different ways. Um. And obviously, if people from another country move into your country, these foreigners have completely different values than what you're used, there's going to be a problem (OU, Interview, MMC4, Stockholm).

Within Extract 8, there is an important delineation between ethnic nationalism and nationalism, and OU relates this to the values of the country. Sweden is positioned as a good, safe country, but importantly, and in line with populist re-bordering narratives, this is placed in the past tense.

### 7.3.3. Extract 9—We Do Not Want These People in Our Country

**First Author:** So, how do you respond to Guterres?

**OU:** The problem now since the world is more connected, yeah, there's internet and all that and there are so many cheap flights to travel around so people travel more. People have more contact with people abroad. So, I guess people know, people are more conscious about other cultures. (First author: yeah) And have stronger opinions about other cultures. So how do you say? So, it's understandable that while the world is sort of expanding and becoming more connected, there will be more nationalism involved because people see more clearly that that country is not good. We do not want those people in our country become racist, nationalistic. It's understandable. I'm not saying it's acceptable but it's understandable. And, and yeah, and there's the multilateralism is under fire precisely when we need it most. I can agree there because we should be helping each other. We should be helping these countries that have problems to become better so to speak so that their people will not want to leave but instead people become very, how do you say? Hostile not wanting, wanting to shut these people out instead. Like we don't want them in our country. Yeah, okay, so if we don't want them there how can we help them? But then again, we need to help our own country first before we can help others so back to that point. Ah, I don't know if that's a good answer for the question there. It's hard (OU, Interview, MMC4, Stockholm).

OU points to the dilemmatic nature of multilateral cooperation when discussing refugee-related migration. Using a feature of the dialogical self, she takes up and co-authors the voice of a xenophobic nationalist: 'We do not want these people in our country'. She simultaneously distances herself from this view but also advocates for it. OU's response challenges us/them accounts of the psychological processes involved in populist re-bordering; her narrative of multilateralism remains continually in a dialogue with xenophobic nationalism (Mahendran 2018).

### 7.3.4. Extract 10—Hate Grows in a Small, Connected World

**OU:** I can agree, since as the world is becoming more connected and smaller, so to speak, people travel more, move more, people all over the world become more conscious about other cultures. This creates sort of problems, since they will then see clearer than before, the differences between each other, and from that more hate grows. Instead of wanting to help poorer, more underdeveloped countries, we want to shut them out, refusing to let them in to our countries, since they will create problems in our home (OU, MMC4, statement to Guterres on IWMT).

For OU, within her fearful narrative, the world becoming smaller, more connected, is not the basis of solidarity; rather, it creates an ease with which people can connect and 'see differences'. Yet, as we found across the 23 interviews, alongside these fears exists an alternative version of the world—a more ideal world. There is both love and hate within OU's narratives and positioning on bordering and multilateralism.

7.3.5. Extract 11—In an Ideal World We Would Move for Good Reasons: Love

> **First author:** Do you want to change your choices? You can, reconsider anything…
>
> **OU:** Which world would I like to have?
>
> **First Author:** Yeah, which one, which is the one that you would work with?
>
> **OU:** I mean to start mapping. It's easy to see here, the one here.
>
> **First Author:** Yeah, so you'd stick with the bordered world?
>
> **OU:** Yeah (First Author: Safe borders?) It's easy to see. (Chuckles) I mean I like the one without borders better. It's nicer. It looks nicer. It's natural but it's easier to see to here from the connecting countries.
>
> **First Author:** Which is the world you would like to live in? (explains option to re-border map or choose different map a second time).
>
> **OU:** Oh okay. In the ideal world if all countries were good then I wouldn't have any borders and I would let it be that way because if all countries were good and safe and had good social security systems then no one would feel the need to flee from the countries because of their regime or you know because of the governments screwing them up. (First author: Yeah) Then people would only move because of better reasons. Maybe they find a partner from another country or maybe they a get a job somewhere else and that's all right I think. And for those reasons, people shouldn't have problems moving.
>
> **First Author:** Yeah, so you would free it up?
>
> **OU:** Yeah.
>
> **First Author:** No borders.
>
> **OU:** No borders, no.
>
> **First Author:** End up with a borderless world (OU, Interview, MMC4, Stockholm).

OU decides finally on the borderless world, travelling a great deal of distance within the study. She creates a good reason for moving. The key here is the extent to which she relates the ideals of multilateralism to migration, the movement of people. This is, of course, partly to do with the parameters of the study, which has asked her to answer questions on her own mobility. She had moved away from Sweden and met someone and then returned with him and settled back into Sweden. This, in Extract 12, becomes the alternative basis of movement for human beings to move for work, love or 'simple curiosity'.

7.3.6. Extract 12—A World without Wars

> **OU:** If the world was an ideal place, without wars, hungers or politicians only looking after themselves instead of the people, then we wouldn't need borders. People would feel the need to move away from their home, only if they wanted to, for example if they met a partner from another country, got a job somewhere else, or simply were curious about another country (OU, MMC4, Stockholm: Final Statement on IWMT about her choice of Google Earth map of the world).

Several articles within this Special Issue point to the possibilities of alternative futures (Andrews et al. 2023), and OU's final statement illustrated the dialogical capacity to imagine possible future worlds when engaged in scientific studies that are designed with these temporalities in mind.

## 8. Discussion

This article contributes to building a bridge between *reified* understandings of multilateralism used by the UN and its General Secretary and *consensual* public understandings of multilateralism. It proposes that public understanding of nationalism and transnationalism within the context of populist re-bordering is foundational to building this bridge.

Within this article, we focused on three cases where, despite different degrees of migration–mobility ranging from YD, a citizen with generational non-mobility (MMC1), to OU, who lived abroad and returned to Sweden (MMC4), and an internal migrant PR who moved from England to Scotland (MMC6), all participants drew on differing political narratives to express an ideal of a border-free world.

In the case of participants with higher mobility, MMC7–MMC10, all participants did not put any lines on the world, fully explored in previous studies and dialogical analyses (Mahendran 2017; Mahendran et al. 2023). This ideal may well be due to the sample of participants along the Migration–Mobility Continuum or indeed be an artefact of our methods. We are aware that the design of the study could create a social desirability effect to not place borders. Of analytical interest to those exploring narratives of resistance is that participants who did place borders (half the participants) did so *whilst* simultaneously holding another *I*-position, that of someone who believed that, in an ideal world, we would not have any borders. The capacity of dialogical citizens to hold a variety of contradictory and complementary positions is a key dimension to studies that work with the idea of a dialogical self (Zittoun 2017; Mahendran et al. 2022), interplaying their micronarratives with the macro-narratives they are interpolated by (Mahendran et al. 2015; Nieland et al. 2022).

As we show in the dialogical analysis of these three case studies, a key component, the foundational abutment to the bridge, is the articulation of how the ideal of a border-free world that was found throughout our study connects to the public's articulation of multilateralism.

Future studies into public understanding of the multilateralism concept could further explore public sense-making within the (i) social relational and cooperative dimensions of multilateralism and (ii) the relationship between multilateralism and economic productivity. Our analysis showed these to be key dimensions when participants were brought into dialogue with a key paragraph of Guterres' 2018 address. Since beginning this article, Finland joined NATO on 4 April 2023, and Sweden's application is no longer blocked by Türkiye. Multilateralism is growing, though that multilateralism since the Russian invasion of Ukraine is potentially aligned along different polarities beyond the UN ideals. It is likely that, post-2022, the public will articulate new narratives around multilateralism. In recent years, there has been much talk of *inclusive multilateralism*, which recognises the extent to which the current discussion on multilateralism does not appear to be aimed at citizens but, rather, is focused on political actors within governments. We found that citizens, when invited to dialogue with Guterres, are not lost for words; rather, they draw on two key social representations relating to a cooperative world and a conflictual world (Staerklé et al. 2011; Mahendran et al. 2023). Equally, when articulating their social representation of cooperation, the basis of the alliance between countries may not be equal statuses but involve newly imagined neocolonial hierarchies or may involve a form of social multilateralism at the level of citizens rather than multilateralism between states.

The critical question becomes how can the political actors involved in articulating reified state actor accounts of multilateralism make a connection between such social representations of cooperation and conflict and the public's ideals about a border-free world that we have explored in this study. Making this connection is the central challenge, we propose, if we are to address Guterres' complaint that multilateralism is under fire precisely when we need it the most. Further articulation of the ideal of a border-free world involves asking two research questions. First, why does such an ideal exist—what specific public narratives and social representations inform this ideal? Many critical migration scholars have argued against such an ideal, and OU argued that a smaller borderless word would increase hate as people became increasingly aware of their differences. Second, within the context of new nationalism (Eger and Valdez 2015; Korkut 2020; Bitonti et al. 2022) and decolonial considerations, as well as a loud, insistent populist re-bordering that characterises the European context of this study, how does the existing dialogue on multilateralism become engaged with this ideal?

Returning to the key themes of this Special Issue, the participants who express this ideal offer cooperative alternative futures before the pandemic began and before canonical accounts of togetherness within the risk of narratives of resilience (Müller and Tuitjert 2022; van Uden and van Houtum 2020), public-level cooperative cross-border ideals may well have found expression in the everyday lived practices of solidarity demonstrated during the pandemic and have the potential to be sustained during the post-pandemic recovery. But the hierarchical basis of such cooperation-shared identities remains a crucial question (English and Mahendran 2021).

**9. Conclusions**

Political and social psychologists, political scientists, narrative studies and, more generally, the social sciences are often preoccupied with xenophobic nationalism and the rise of nationalistic forms of populism and are actively engaged in studying them. We propose that, to understand the protectionist tensions that are likely to increase during the austerities of the post-pandemic recovery, rather than more studies into populism, such scholars need to consider directly investigating the public's narratives of and engagement with multilateralism. A valuable new departure for such scientists involves how this relates to their ideals about the world, their worldviews, their understandings of global orders and, increasingly, their *pandemicality*, in the sense of their dialogical capacity to think and act according to *planet views*.

**Author Contributions:** Conceptualisation, K.M.; Formal analysis, K.M., A.E. and S.N.; Writing—original draft, K.M., A.E. and S.N.; Writing—review and editing, K.M., A.E. and S.N. All authors have read and agreed to the published version of the manuscript.

**Funding:** This research was funded by the Open University, Faculty of Arts and Social Sciences, Strategic Research Innovation Fund.

**Institutional Review Board Statement:** The Citizen Worldviews Mapping Project was conducted in accordance with BPS Code of Human Research Ethics and was reviewed by, and received a favourable opinion from, the Open University Human Research Ethics Committee (HREC), reference number: HREC/3280/Mahendran.

**Informed Consent Statement:** All subjects gave their informed consent for inclusion before they participated in the study. The study was conducted in accordance with the BPS details.

**Data Availability Statement:** The data are not publicly available due to privacy issues.

**Acknowledgments:** The authors would like to thank the two guest editors, Molly Andrews and Paul Nesbitt-Larking, who read several versions of this article at all stages. We would also like to thank the two anonymous peer reviewers for their thoughtful and constructive feedback. Ulla Boehme, Sara Ferlander and Hans Ingvar-Roth for facilitating the Swedish fieldwork; and finally, we thank the participants who participated in the study.

**Conflicts of Interest:** The authors declare no conflict of interest.

**Notes**

[1] Folkhemmet was originally a unifying social cohesion concept of the people's home used within the social democratic model but was successfully reappropriated by Sverigedemokraterna (Sweden Democrats) (see author ref).

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
