# Peer review of "Multilateralism under Fire: How Public Narratives of Multilateralism and Ideals of a Border-Free World Repudiate the Populist Re-Bordering Narrative"

_socsci, doi:10.3390/socsci12100566_

Round 1
Reviewer 1 Report
The manuscript “Multilateralism under fire: How public narratives of multilateralism and ideals of a border-free world repudiate the populist re-bordering narrative” focuses on a relevant social theme and fits the scope of the journal Social Sciences and of this particular special issue. The text is well-written and presents a clear rationale regarding the authors’ aims and methodological choices. Besides, the authors include relevant and recent references in the Introduction. However, the Discussion section needs some improvement: several parts of this section could be relocated to the Conclusion and at least two paragraphs could be added to the Discussion, in order to further explain the results, in dialogue with the literature cited in the Introduction (which was not sufficiently incorporated into the discussion of the results).
Furthermore, after reading the manuscript, I was able to find some of the authors’ previous work and I noticed that several paragraphs of this text are really similar to some parts of at least one of the articles already published by the authors. Therefore, it is important that the authors check these similarities and make appropriate changes in this manuscript if needed to avoid incurring self-plagiarism.
Apart from these two key points (improving the discussion and watching out for self-plagiarism), I have only other minor observations regarding specific aspects of the manuscript:
-L. 29: “Covid-19 coronavirus pandemic” – there is no need to repeat the word “coronavirus” after “COVID”, which means Coronavirus disease.
-L. 49: “alternatives futures” - alternative futures
Author Response
Authors revised the comments accordingly.
Reviewer 2 Report
The article under review presents an original, innovative and well-contextualized analysis of the relevance of the concept of multilateralism.
I agree with the theoretical mapping. I would just like to make it clear that it would have been very relevant for the text to discuss multilateralism from an interdisciplinary perspective, namely with the work of some researchers who have made an unparalleled contribution to the historical, cultural and sociological awareness of this issue.
As noted at the beginning of the article: “as political psychologists are concerned with the dialogue between citizens and their governments…”. With this in mind, resorting to other approaches and contributions from postcolonial studies, African studies and memory studies would have been an excellent choice for a better theoretical framework. To point out some works, that I believe will enrich this article:
-
https://www.tandfonline.com/doi/epdf/10.1080/0031322X.2018.1433004?needAccess=true&role=button; (Dan Stone, 2018);
-
Peter Gatrell, Refugees—What’s Wrong with History?, Journal of Refugee Studies, Volume 30, Issue 2, June 2017, Pages 170–189, https://doi.org/10.1093/jrs/few013
-
Knudsen, B. T., Oldfield, J. R., Buettner, E. & Zabynuan, E. (2021). Echoes of coloniality: New perspectives on Decolonizing European Heritage. New York: Routledge.
-
Koegler, C., Kumar, P. M. & e Tronicke, M. (2020). The colonial remains of Brexit: Empire nostalgia and narcissistic nationalism. Journal of Postcolonial Writing, 56:5, 585-592. DOI: 10.1080/17449855.2020.1818440.
-
Lowe, L. (2015). History Hesitant. Social Text, Vol.33, no.4, December,85-107. https://doi.org/10.1215/01642472-3315790
-
Trafford, J. (2021). The Empire at Home Internal Colonies and the End of Britain. London: Pluto Press.
-
El-Enany, N. (2020). Bordering Britain, Law, Race and Empire, Manchester: Manchester University Press.
-
Essed, P. &, Nimako, K. (2006). Designs and (Co)Incidents: Cultures of Scholarship and Public Policy on Immigrants/ Minorities in the Netherlands. International Journal of Comparative Sociology, 47(3-4), 281–312. https://DOI.org/10.1177/0020715206065784
Author Response
Authors revised comments accordingly